# Root traits and belowground herbivores relate to plant–soil feedback variation among congeners

Rutger A. Wilschut [1,2], Wim H. van der Putten [1,2], Paolina Garbeva[3], Paula Harkes [2], Wouter Konings[1], Purva Kulkarni[3], Henk Martens[1] & Stefan Geisen [1]

Plant–soil feedbacks contribute to vegetation dynamics by species-specific interactions between plants and soil biota. Variation in plant–soil feedbacks can be predicted by root traits, successional position, and plant nativeness. However, it is unknown whether closely related plant species develop more similar plant–soil feedbacks than more distantly related species. Where previous comparisons included plant species from distant phylogenetic positions, we studied plant–soil feedbacks of congeneric species. Using eight intracontinentally range-expanding and native *Geranium* species, we tested relations between phylogenetic distances, chemical and structural root traits, root microbiomes, and plant–soil feedbacks. We show that root chemistry and specific root length better predict bacterial and fungal community composition than phylogenetic distance. Negative plant–soil feedback strength correlates with root-feeding nematode numbers, whereas microbiome dissimilarity, nativeness, or phylogeny does not predict plant–soil feedbacks. We conclude that root microbiome variation among congeners is best explained by root traits, and that root-feeding nematode abundances predict plant–soil feedbacks.

---

[1] Laboratory of Nematology, Wageningen University, PO Box 8123 , Droevendaalsesteeg, 16700 ES Wageningen, The Netherlands. [2] Department of Microbial Ecology, Netherlands Institute of Ecology, PO Box 50 , Droevendaalsesteeg, 106700 AB Wageningen, The Netherlands. [3] Department of Terrestrial Ecology, Netherlands Institute of Ecology, PO Box 50 , Droevendaalsesteeg, 106700 AB Wageningen, The Netherlands. Correspondence and requests for materials should be addressed to R.A.W. (email: r.wilschut@nioo.knaw.nl)

It is increasingly acknowledged that soil biota contribute to the control of plant species diversity by causing positive or negative plant–soil feedbacks[1–4]. These feedback effects are caused by the accumulation of antagonistic and mutualistic-symbiotic soil organisms by plants, which thereby reduce or promote themselves, their neighbors, or their successors[3,5]. Plant species differentially shape soil communities by a number of mechanisms, including the production of unique combinations of root exudates and volatiles[6,7]. However, the resulting changes in soil communities that feedback to plant performance often remain unknown and we therefore largely lack an understanding of the exact biotic drivers of plant–soil feedbacks.

Soil communities in the rhizospheres of plant species are expected to vary with phylogenetic distance between plant species, as phylogenetically closely related plant species are more likely to share structural and chemical traits than distantly related species[8,9]. In turn, closely related plant species might also be similarly affected by their conditioned soil communities, as they often have comparable natural enemies[10,11] and mutualistic symbionts[12,13]. As plant–soil feedbacks are net effects of positive and negative interactions between plants and soil biota[14], it may be assumed that plant–soil feedback is, at least in part, phylogenetically determined.

So far, studies have found mixed support for a predictive relationship between phylogeny and plant–soil feedbacks[15–17]. In most of these studies, hetero-specific plant–soil feedback was examined: the authors tested whether the phylogenetic distance between conditioning plant species predicts the feedback to other plant species[16,18–20]. The effect of plant relatedness on plant–soil feedback effects on conspecifics has so far only been tested along a wide phylogenetic gradient[17]. However, in these types of comparisons, deeply conserved traits, which show variation on higher taxonomic levels such as plant families, will likely determine the presence of a phylogenetic signal. Therefore, such studies may not be suitable to test whether phylogenetic distances can be used to understand variation in plant–soil feedbacks among closely related congeneric species. Here we specifically determined whether closely related plant species have a more comparable plant–soil feedback than more distantly related plant species by using plant species from the same genus. We examined how rhizosphere microbiome variation related to phylogenetic distance within this genus, as the microbiome is expected to underlie plant–soil feedback effects[12]. So far, only few studies have examined rhizosphere microbiome variation in relation to plant phylogeny[21]. Instead, most studies only examined single groups of rhizosphere organisms[22,23], whereas we included the full soil rhizosphere community.

Plant–soil feedbacks can strongly differ between native plant species and introduced exotics, which may contribute to the success of non-native plant species in their new range[4,24,25]. Reduced negative plant–soil feedbacks of non-native plant species in their new range is not found for invasive exotics[25] but also for plant species that expand their range within the same continent, as a consequence of climate warming[26–28]. However, the biotic interactions potentially underlying plant–soil feedback differences between non-natives and natives may be explained by differences in root chemistry[29]. Therefore, variation in plant–soil feedback outcomes between non-natives and natives might be predicted with phylogenetic distance, when root traits underlying these biotic interactions are phylogenetically conserved[8,9]. However, it is unknown whether plant origin, in addition to phylogenetic distance, may influence plant–soil interactions of non-native plant species in their new range. For example, introduced exotics or range-expanding plant species can become released from natural soil-borne pathogens, even in the presence of a closely related plant species[30,31].

Here we test the hypothesis that differences in rhizosphere community composition and plant–soil feedback among range-expanding and congeneric native plant species, and the traits underlying this variation, are explained by their phylogenetic distances. We test our hypothesis using eight congeneric *Geranium* species that all occur in north-western Europe. Four of these species are native, whereas the other four recently have become established, most likely as a consequence of recent climate warming. We are able to test plant origin effects irrespective of phylogenetic distance, because native and range-expanding plant species are not phylogenetically clustered (Supplementary Fig. 1). We combine a plant–soil feedback experiment with sequencing of prokaryotic and eukaryotic communities conditioned by the different plant species, and perform analyses of root chemical profiles and structural root traits. Thereby, we relate plant–soil feedback effects to variations in soil community composition and belowground plant traits, and determine whether closely related plant species have a more comparable plant–soil feedback than more distantly related plant species.

## Results

**Rhizosphere community composition.** The conditioned prokaryotic (16S rRNA gene reads) and eukaryotic (18S rRNA gene reads) rhizosphere communities varied significantly between the eight plant species (Fig. 1a, b and Supplementary Table 1). The composition of the three distinct taxonomic groups composing the 18s rDNA, fungi, protists, and nematodes also differed between the plant species (Supplementary Table 1 and Supplementary Fig. 2). Compositional differences in any of these communities were not explained by plant origin, indicating that soil communities in general were not differently conditioned by natives than by related range expanders (Supplementary Table 1). Between-species dissimilarity of the full prokaryotic and eukaryotic rhizosphere communities did not correlate with the phylogenetic distance between the plant species (Fig. 1c and Supplementary Table 1). However, distantly related plant species had more dissimilar fungal communities than closely related plant species (Mantel test: $r = 0.39$, $p < 0.05$; Fig. 1d). Rarefaction curves showed that we did not fully sample all biodiversity, as no plateau in the rarefaction curves was reached (Supplementary Fig. 3). This indicates that we likely have missed members of the rare biosphere. However, the intend of our study was to assess the importance of the major diversity and functional groups within the studied soil biota.

Further analyses showed that variation in fungal community composition correlated with differences in specific root length (Mantel test: $r = 0.41$, $p < 0.05$; Fig. 2, Supplementary Fig. 4), whereas differences in bacterial community composition tended to correspond with variation in root metabolic profiles (Mantel test: $r = 0.52$, $p = 0.08$; Fig. 2, Supplementary Figure 4). Average root diameter did not explain variation in any of the groups in the rhizosphere community (Supplementary Fig. 4). Between-species variation in root chemical profiles could not be explained by phylogenetic distance, whereas differences in specific root length and average root diameter marginally significantly correlated with phylogenetic distance (Supplementary Fig. 5). Between-plant species, protist, and nematode communities co-varied with the community composition of bacteria, whereas the composition of nematode communities also co-varied with the composition of fungal communities (Fig. 2 and Supplementary Fig. 6).

A structural equation model was constructed based on the correlational links between root traits and rhizosphere community dissimilarity (Fig. 2). This model confirmed the effect of root chemical profile and specific root length on the bacterial and fungal community dissimilarity, respectively (Supplementary Fig. 7). Moreover, it showed that protist community dissimilarity

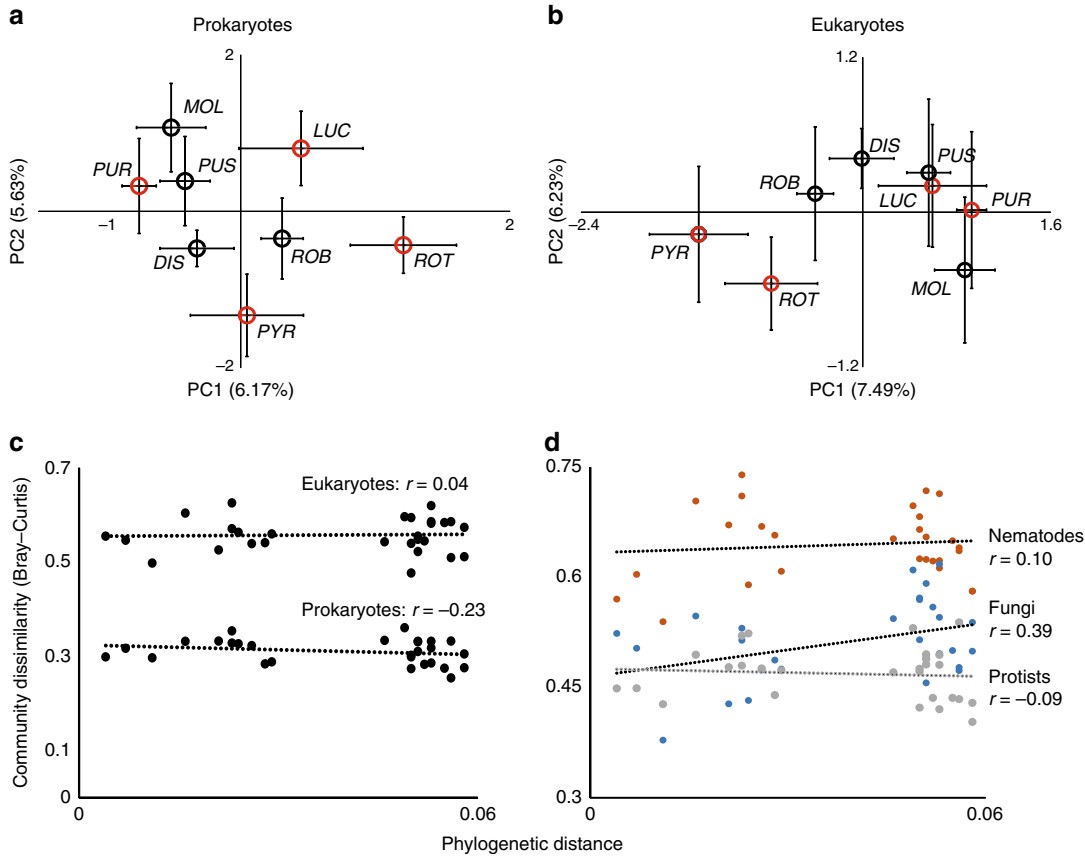

**Fig. 1** Community composition of prokaryotes and eukaryotes. Compositional variation in rhizosphere communities of **a** prokaryotes (16S rRNA gene reads) and **b** eukaryotes (18S rRNA gene reads) among native (black dots: *G. dissectum* (DIS), *G. molle* (MOL), *G. pusillum* (PUS), and *G. robertianum* (ROB)), and range-expanding plant species (red dots: *G. lucidum* (LUC), *G. pyrenaicum* (PYR), *G. purpureum* (PUR), and *G. rotundifolium* (ROT)), based on five independent replicate soils. Error bars represent SEs of PCA coordinates. Correlations of pairwise phylogenetic distance with community dissimilarity of prokaryotes and eukaryotes (**c**), and eukaryotic groups nematodes (orange), fungi (blue), and protists (gray) (**d**). The Pearson's coefficient *r* for each correlation is shown

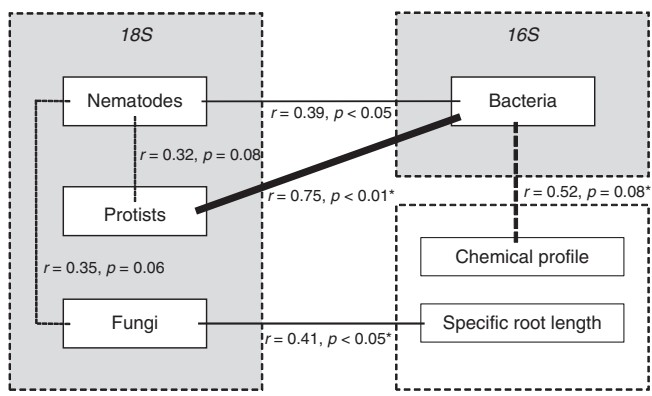

**Fig. 2** Correlational links in the rhizosphere. Overview of correlational links (all correlations with *p* < 0.1 are shown) between rhizosphere community dissimilarities of bacteria, nematodes, protists, and fungi (all based on OTUs; 4–5 replicates per species) and between root trait variation (root chemical profile (4 replicates per species), specific root length (3 replicates per species)), and dissimilarities of rhizosphere communities. Significant correlations (based on Mantel tests) are depicted with solid lines, whereas trends (*p* > 0.05, < 0.10) are depicted with dashed lines. Line thickness represents the relative strength of the correlational link based on *r*-values. Links that are significant based on a structural equation model (Supplementary Fig. 7) are represented with asterisks (*). Source data are provided as a Source Data file

could be well predicted by bacterial community dissimilarity (Supplementary Fig. 7).

**Plant–soil feedback**. There was an overall effect of soil conditioning (general linear model: $F_{1,60} = 435.6$, $p < 0.001$) on plant performance in the feedback phase and all species grew equally poor in soil conditioned by their conspecifics (Fig. 3a). However, the species differed profoundly in their proportional loss of biomass in response to soil conditioning and thus in their plant–soil feedback responses (general linear model: conditioning × species: $F_{7,60} = 6.20$, $p < 0.001$). On average, range-expanding plant species responded more negatively to soil conditioning than natives (general linear model: contrast range-expanders natives: $F = 13.81$, $p < 0.001$), which was mainly due to the greater biomass of the range-expander *Geranium purpureum* in unconditioned soils and its relatively low biomass in conditioned soils (Fig. 3a). Pairwise comparisons of plant–soil feedback strength did not reveal that plant–soil feedback is phylogenetically determined in this group of plant species (Mantel test: $r = -0.03$, $p = 0.49$; Fig. 3b).

Plant–soil feedback variation was neither correlated with dissimilarity in complete 16S and 18S communities, nor with the dissimilarity in fungal, nematode, or protist communities (Supplementary Fig. 8). These results motivated us to explore the relationship between specific organismal groups of soil biota and the strength of plant–soil feedbacks. We tested the correlation between the relative abundances of potential mutualistic, arbuscular mycorrhizal fungi (AMF), and eukaryotic plant pathogens and

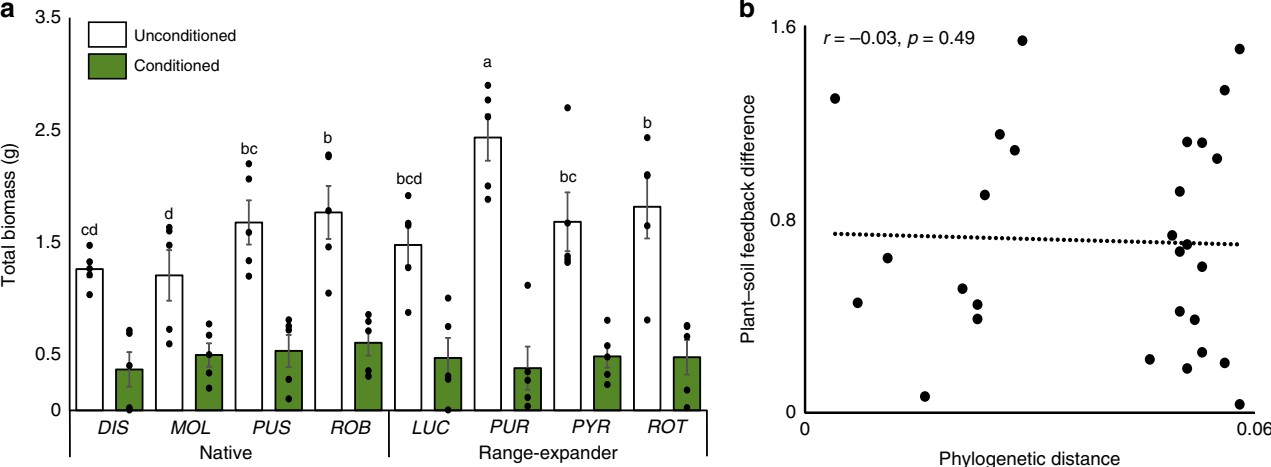

**Fig. 3** Plant–soil feedback. **a** Plant biomass of eight *Geranium* species in soils conditioned by conspecifics (green) or in unconditioned soils (white) ($N = 5$ per species). Native plant species are *G. dissectum* (DIS), *G. molle* (MOL), *G. pusillum* (PUS), and *G. robertianum* (ROB) and range-expanding plant species are *G. lucidum* (LUC), *G. pyrenaicum* (PYR), *G. purpureum* (PUR), and *G. rotundifolium* (ROT). Bars and whiskers represent average biomass ± SEs as examined using a general linear model. Small letters show post-hoc test results between plant species in unconditioned soils. **b** Absolute between-species differences in average plant–soil feedback (Ln(biomass$_{conditioned}$/biomass$_{control}$)) do not correlate with pairwise phylogenetic distance as examined using a Mantel test. Source data are provided as a Source Data file

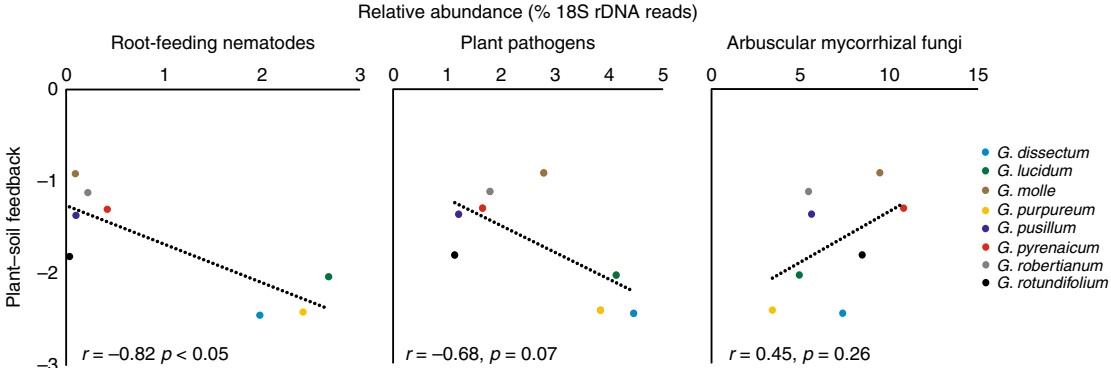

**Fig. 4** Predictors of plant–soil feedback. Correlations between plant–soil feedback (Ln(biomass$_{conditioned}$/biomass$_{control}$); average of five independent replicate soils) of eight *Geranium* species and the relative abundances (% 18S rRNA gene reads) of root-feeding nematodes, plant pathogens (see Methods), and arbuscular mycorrhizal fungi (AMF) in the conditioned soils. Statistical differences were examined using a Pearson's correlation test. Pearson's correlation coefficients $r$ and $p$-values of Pearson's correlation tests are shown. Source data are provided as a Source Data file

root-feeding nematodes (see methods). Relative abundances of root-feeding nematodes correlated with plant–soil feedback strength (Mantel test: $r = 0.82$, $p < 0.05$; Fig. 4, Supplementary Fig. 9). There was a weak trend that also the relative abundance of plant pathogens correlated with the strength of negative plant–soil feedback, whereas there was no correlation between plant–soil feedback and AMF abundance (Fig. 4 and Supplementary Fig. 9). Relative abundances of root-feeding nematodes did not correlate with plant biomass in the conditioning phase, indicating that root traits other than biomass determine the accumulation of these organisms (Supplementary Fig. 10).

Of all root-feeding nematode taxa, the genus *Meloidogyne* was most abundant in the conditioned soils (68% of root-feeding nematodes reads per sample). To test whether plant–soil feedback is also related to absolute root-feeding nematode abundance, we correlated plant–soil feedback with the reproduction of the widespread species *Meloidogyne hapla* on all eight plant species, which was examined in a separate experiment. *Meloidogyne hapla* numbers depended on plant species identity (generalized linear model: $X^2 = 134.38$, $p < 0.0001$; Fig. 5a) and plant species that

developed the most negative plant–soil feedbacks indeed were the best hosts for *Meloidogyne* ($r = -0.72$, $p < 0.05$; Fig. 5b).

## Discussion

Our results show that variation in rhizosphere community composition among eight congeneric native and climate warming-driven range-expanding plant species could be explained most strongly by variation in their root chemical profiles and morphological root traits. Only differences in fungal community composition could, at least in part, be explained by phylogenetic distance, possibly underlain by the phylogenetic signal in specific root length. Communities of fungi varied with specific root length, which is possibly explained by plant interactions with root-associated fungi, such as AMF[32]. The link between root chemistry and bacterial community composition is in line with previous research[33]. Interestingly, there was evident co-variation in the composition of the different rhizosphere groups among plant species, especially between bacteria and protists. This is likely explained by feeding relationships between

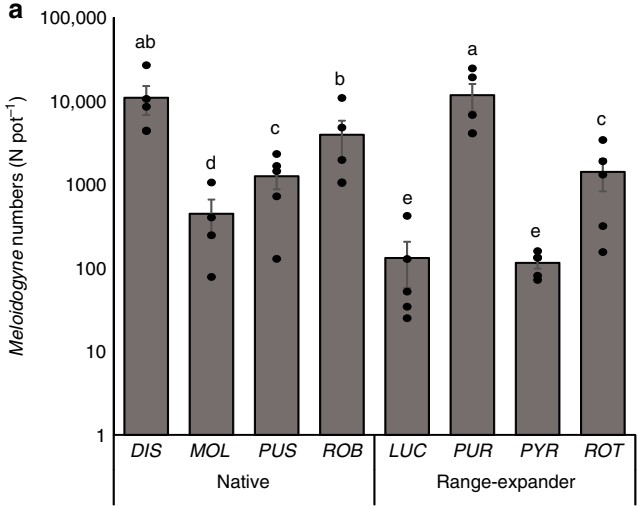

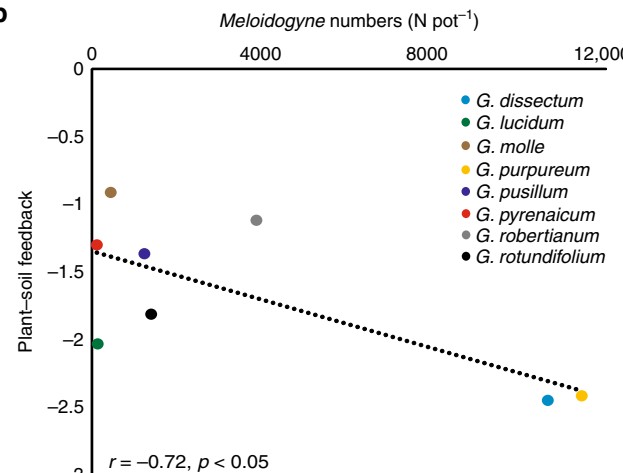

**Fig. 5** Nematode reproduction. **a** Abundances of root-feeding nematode *M. hapla* on native (black) and range-expanding (red) *Geranium* species in a nematode reproduction experiment ($N = 5$ per species). Bars represent average nematode numbers ± SEs. Letters indicate the significant differences based on negative binomial GLM and post-hoc Wald's tests. Note that the y-axis has a logarithmic scale. **b** Correlation between *Meloidogyne* abundances and plant–soil feedback (ln(biomass$_{conditioned}$/ biomass$_{control}$). A Pearson's correlation test was applied to evaluate the correlation. Pearson's correlation coefficients $r$ and $p$-values of Pearson's correlation tests are shown. Source data are provided as a Source Data file.

these two groups[34,35] and suggests that protist feed species-specifically on bacteria.

Variation in plant–soil feedbacks could not be directly linked to the composition of the rhizosphere community, but rather to the abundances of groups of antagonistic soil organisms, especially root-feeding nematodes. These nematode communities were dominated by *Meloidogyne* species, which are among the most detrimental nematode pests[36] and plant–soil feedback correlated to the reproduction of a generalist root knot nematode, *M. hapla*. This suggests that generalistic belowground natural enemies drive plant–soil feedbacks in this study rather than specialists that are expected to underlie the Janzen–Connell dynamics, which have been shown to correlate with phylogenetic distance[37].

Although root-feeding nematodes have been widely acknowledged as major agricultural pests[38], we here show that they also likely affect plant performance in natural systems[39]. Further research might consider interactions between plant species and

soil type, increased numbers of replicates to account for the variation, and inoculation trials in order to experimentally verify the role of plant-feeding nematodes in plant growth reduction. Furthermore, deeper sequencing, the use of nematode-specific primers, or longer-read sequencing in future studies should be considered to increase taxon sampling and the taxonomic resolution focusing on root-feeding nematodes, as well as other soil-borne pathogens, in order to potentially identify taxa that are specifically hosted by the different plant species.

Interestingly, the highest relative abundances of potential plant pathogens were found in the plants that hosted highest nematode numbers, and that experienced strongest negative plant–soil feedbacks. This suggests that these plants are generally less well defended against a wide array of natural enemies in the soil. Apart from the biotic actors, the observed negative plant–soil feedbacks may to some degree have been caused by differences in nutrient availability between the control and conditioned pots. Although we added Hoagland solution to compensate for the lack of macro-nutrients, it cannot be excluded that plants in the feedback phase still suffered from shortage of certain micronutrients.

Our experiment revealed no plant origin effect on rhizosphere community composition, unlike previously assumed[40]. In contrast to previous work[26], we found that range-expanding plant species on average developed a more negative plant–soil feedback than native species. However, the plant–soil feedback effect in the present study was likely mainly driven by one single range-expanding plant species (*G. purpureum*; Fig. 3a) and plant–soil feedback outcomes differed within both native and range-expanding plant species. Moreover, the previous studies showed plant–soil feedback values across plant families, whereas the present comparison was made within only one plant genus. Finally, information on the plant species used in the present study suggests that the negative plant–soil feedbacks of some of the range-expanding *Geranium* species most likely have not hampered their successful establishment in the new range[41].

Phylogenetic distance has been successfully used as a measure of ecological (dis)similarity and as a predictor of biotic interaction outcomes in studies that included plant species from multiple families[11,17]. It is likely to be that ecological differences in such studies may be determined by deeply conserved traits that can vary between families. Our study shows that among a group of congenerics, even the most closely related species (e.g., *Geranium robertianum* and *G. purpureum*) can have stronger differences in rhizosphere communities and plant–soil feedback effects than less closely related species. Therefore, our study challenges the use of phylogenetic distance as a measure to explain plant–soil inter-action patterns of closely related plant species. Instead, we show that non-phylogenetically conserved root traits may be more effective to predict plant–soil interactions. Thus, in order to fully understand plant–soil feedback variation among more closely and more distantly related plant species, a combination of phylo-genetically conserved traits and unconserved traits is needed.

Our results raise the question under which conditions phylo-genetic distance will explain variations in plant–soil feedbacks among closely related species. The examined *Geranium* species differ in their preferred abiotic conditions[42] (Supplementary Table 4) and do not all co-occur under the same field conditions. This suggests that their root traits have been selected in the presence of different soil communities, which may co-vary with abiotic soil conditions[43,44]. Congeneric plant species that co-occur under the same abiotic soil conditions may face more similar selection pressures[11] and in turn may have root traits that more strongly resemble their phylogenetic history. Moreover, plant species from different genera with comparable root traits[45] or with similar life histories[3] might have more comparable

plant–soil feedback effects than closely related plant species from contrasting environments, as have been compared in our study.

We conclude that within the same plant genus, closely related plant species did not have a more comparable microbiomes and plant–soil feedbacks than more distantly related plant species. Enemy release is likely not underlying the success of *Geranium* species to expand their range as the performance of native and range-expanding plant species was similar. Root traits were the best predictors of rhizosphere community composition and root-feeding nematode numbers—and not rhizosphere community dissimilarity—best predicted the strength of plant–soil feedback.

## Methods

**Plant species and germination.** We collected seeds of native *Geranium* species *G. dissectum* L., *G. molle* L., *G. pusillum* L., and *G. robertianum* L., and range-expanding species *G. lucidum* L., *G. purpureum* Vill., *G. pyrenaicum* Burm.f., and *G. rotundifolium* L. from natural populations in The Netherlands (Supplementary Table 2). We identified Ellenberg indicator values for each of those species (Supplementary Table 3). All natives naturally occur in the Netherlands, whereas the range-expanders established populations in north-western Europe in the late twentieth century (*G. lucidum* and *G. purpureum*) or were already present in restricted areas and strongly expanded their range in the last decades of the twentieth century (The Netherlands: http://verspreidingsatlas.nl[41], United Kingdom: https://www.brc.ac.uk/plantatlas). For each experiment, seeds were surface-sterilized by washing them for 3 min in a 10% bleach solution, followed by rinsing with demineralized water, after which they were germinated on glass beads.

**Phylogeny reconstruction.** We concatenated three barcoding regions commonly used to infer plant phylogenies: the genes *rbcL*[46] and *trnL*[47], as well as the intergeneric spacer trnL-trnF[48]. Due to multiple sequences for rbcL present in GenBank, we decided to re-amplify the *rbcL* gene for all plants used in our experiment. For this, root DNA was extracted from all *Geranium* species using the PowerSoil DNA Isolation kit (Qiagen, USA), which was adjusted by using iron beads to increase physical impact. We amplified the large chain of the ribulose bisphosphate carboxylase (rbcL) using the primers 1F[46] and the newly designed primer 1361rMod (5′-TATCCG-TAAGGCTTGCAAGTGGAGT-3′) modified from a previously described primer[49], with PCR cycling conditions as follows: initiation for 5 min at 95 °C, followed by 30 cycles of 30 s at 95 °C, 1 min at 59 °C, and 75 s at 72° with a final elongation for 5 min at 72 °C). DNA sequencing was performed by LGC Limited (Middlesex, UK). Obtained sequence chromatograms were manually curated in Chromas Lite v 2.11 (http://chromas-lite.software.informer.com/2.1/; Technelysium, Queensland, Australia). Curated sequences were uploaded to NCBI under the accession numbers MK542498-MK542505. Sequences of the new *rbcL* gene reads as well as the trnL and the intergeneric spacer trnL-trnF were aligned using MAFFT[50] and visualized in Seaview v4.6.3[51]. Maximum likelihood analyses were run directly in Seaview using PhyML using the generalised time-reversible (GTR) model with four rate categories based on 2251 nucleotide sites. The stability of the branches in the resulting phylogenetic tree was assessed based on 1000 bootstrap replicates. Pairwise phylogenetic distance was then estimated with the ratio of divergent nucleotides.

**Soil-conditioning experiment.** The plant–soil feedback experiment consisted of two phases. For the conditioning phase, we prepared a common background soil by homogenizing sandy clay soil from a former agricultural field (Beneden-Leeuwen, The Netherlands; N51° 53.952, E05° 33.670) with sand, after which it was sterilized using gamma-sterilization (25 KGray, Syngenta bv, Ede, The Netherlands). To establish independent replicates, we collected field soil from five different sites in the same river valley (Supplementary Table 4), each with four different subsamples. These subsamples were pooled, sieved, and homogenized, after which the mixtures were kept separate throughout the experiment as five replicate soils. Per replicate soil, sixteen 2.5 l pots were filled with a mixture of 1.8 kg of sterilized background soil and 200 g of sieved (1 cm) alive field soil. For each replicate soil, eight pots were planted with one of the eight different *Geranium* species, whereas the other pots were left unplanted. All pots were then positioned in a randomized block design with five replicate blocks in a climatized greenhouse (16/8 h light/dark and 20/15 °C). For the next 14 weeks, the pots were watered twice per week and kept at the same soil moisture content (~15%). Thereafter, shoots were clipped, dried, and weighed, whereas roots were washed, dried, and weighed. Soils from each pot were collected and kept separate, and a subsample was stored at −4 °C for DNA extraction.

**Feedback experiment.** In the second phase, soils from each of the first phase pots were individually transferred to 1 liter pots, which were filled with 830 g soil (moisture content ~15%) and put in the same randomized block design as in the first experimental phase. Soils that were conditioned by a plant in the first phase were planted with a seedling of the same species in the second phase. Each pot with unconditioned soil was also planted with one of the eight species. The same watering regime was applied as in the first phase. To compensate for differences in nutrient availability

originating from the conditioning phase, all pots each week received 10 ml of 25% Hoagland solution from the second week onwards, so that all plants had ample available nutrients. After 7 weeks of plant growth, shoots and roots were collected as described above.

**Soil DNA extraction.** For each pot with conditioned soil, DNA was isolated on the principle of the MoBio PowerSoil DNA isolation kit. Briefly, 1 g of soil was added to 3 ml bead solution and 0.24 mL of an SDS-based lysis buffer. Eight iron spheres (ø 3 mm) were added to increase extraction efficiency. Tubes were shaken for 6 min at 580 r.p.m. in a paint shaker (COROB™ SIMPLEShake). After centrifuging, the supernatant was abstracted from humic acids using 0.8 mL of an ammonium aluminum sulfate dodecahydrate solution. A vacuum manifold with a 96-well plate containing a silica membrane (PALL8032) was used in order to purify DNA. DNA quantity was measured with nanodrop.

**16S and 18S rRNA gene amplicon sequencing.** The community structure of prokaryotes (bacteria and archaea) was determined using the prokaryote-wide primers 515F/806R targeting the V4 region of the 16S rRNA gene[52]. The eukaryotic community structure was assessed using the general eukaryotic primers 3NDf[53] and 1132rmod[54] targeting the most variable V4 region of the 18S rRNA gene[55]. All primers were pre-tagged with Illumina adapters, a 12 bp long barcode to allow demultiplexing of the reads after sequencing, a primer linker, and the sequencing primers. PCRs were conducted using in 96-well plates containing 25 µl mixes. For prokaryotes, these mixes included 11.75 µl milliQ water, 10 µl 5 Prime Hot 2.5× mastermix (QuantaBio), 1.25 µl bovine serum albumin (BSA) 4 µg/µl, and 0.5 µl of each of the primers and 1 µl of DNA template. PCR conditions consisted of an initial denaturation step at 94 °C for 5 min followed by 35 cycles of 94 °C for 45 s, 50 °C for 60 s, and 72 °C for 90 s with a final elongation at 72 °C for 10 min. For eukaryotes, these mixes included 15.3 µl milliQ water, 1 µl dNTPs, 1.25 µl BSA 4 µg/µl, 2.8 µl MgCL₂, 0.5 µl of each of the primers, 10 µl buffer with 0.15 µl Taq polymerase (Fast start, Roche), and 1 µl of DNA template. PCR conditions were identical to those targeting prokaryotes with only the annealing temperature increased to 54 °C. All PCRs were performed in duplicates, before quality assessment on 1.5% agarose gel. PCR duplicates were pooled and cleaned using Agencourt AMPure XP magnetic beads (Beckman Coulter). DNA concentrations were assessed with a fragment analyzer (Advanced Analytical), pooled in equimolar ratios, and sent for sequencing to BGI, China.

**Bioinformatics.** The obtained raw 16S and 18S rRNA gene sequence reads were curated in the Hydra pipeline (nioo-knaw/hydra (Version 1.3.3)) implemented in Snakemake[56]; in short, after filtering contaminants and removing barcodes, 16S rDNA reads were merged with the fastq_mergepairs option of vsearch[57], whereas for the 18S rRNA gene data the forward reads were used. Thereafter, for both 16S and 18S rRNA gene reads, VSEARCH was used to cluster all reads into operational taxonomic units (OTUs) using the UPARSE strategy by de-replication followed by sequence-sorting by abundance (singletons were removed) and clustering using the UCLUST smallmem algorithm[58]. Chimeric sequences were removed using UCHIME[59], implemented in VSEARCH. To create an OTU table, all reads were mapped to OTUs using the usearch_global method (VSEARCH). OTUs obtained from 16S rRNA gene sequences were taxonomically assigned by aligning them to the SILVA database[60]; 18S rRNA gene sequences were aligned to the PR2 database[61]. Reference sequences were first trimmed with forward and reverse primer using cutadapt[62]. We constructed rarefaction curves to estimate sampling saturation, separately for bacteria, protists, nematodes, and fungi using Past statistics (Supplementary Figure 3). Before the analyses, we deleted all OTUs present in <25% of the samples. Moreover, we removed samples with fewer than 3000 18S rRNA gene reads from further analyses (see Supplementary Table 5). Although a small number of these removed OTUs had high abundance in some samples, none of them was consistently abundant in any of the treatments. All 16S rRNA gene read samples contained at least 17,000 reads and therefore none was discarded from further analyses. We then recalculated read numbers to relative abundances of the OTUs. OTUs were then manually assigned into the functional groups (Supplementary Table 6), allowing estimates of relative abundances of root-feeding nematodes[63], AMF (Glomeromycota), and plant pathogens (Plasmodiophorida, Oomycetes and *Rhizoctonia* sp.).

**Structural root traits.** For each plant species, three seedlings were grown individually in sterilized soil as described above. After 4 weeks of growth, all plants were stored at 4 °C until root trait analyses. Before this analysis, shoots were clipped and dried at 70 °C until constant weight, whereas root systems were carefully washed. Individual root systems then were fragmented and scanned using an Epson Perfection V850 Pro scanner (Epson America, Inc). Scans were subsequently analyzed using WINRHIZO Pro v.2005b[64] for total root lengths and mean diameters. After scanning, root systems were dried until constant weight and weighed, after which the root/shoot ratio was determined.

**Root chemistry analysis.** For all plant species, four 5-week-old plants were harvested from sterilized soil, after which their root systems were carefully washed. Thereafter, we used Direct Analysis in Real Time (DART) mass spectrometry to determine the root chemical profile of all plant species. The DART mass spectrometry set-up consists of a DART ion source (model DART-SVP, IonSence,

Saugus, USA) coupled with Q Exactive Focus high-resolution mass spectrometer (Thermo Fisher Scientific, San Jose, CA, USA). The mass spectrometer was calibrated before the samples measurements. The Xcalibur software (v.3.0) was used for instrument control and data acquisition. The distance between mass inlet and the DART outlet was kept at ~3 cm. To standardize the measurements, root samples were placed on glass plates and automatically moved (0.4 mm/s) along the ion source. DART settings were as follows: helium as ionizing gas, fixed flow of ~3.5 L/min; gas beam temperature set at 450 °C; grid electrode voltage +350 V. The resolution was set at ultrahigh and a scan rate of 1 Hz was used. The mass spectra were recorded in the m/z range 100–1500 at acquisition rate of 2 spectra/s.

**Mass spectrometry data processing**. The DART-MS spectra were acquired and converted from their respective raw data formats to open-source mzXML file format using MSConvertGUI (64 bit) available from ProteoWizard[65]. For further mass spectral data processing, the open-source software package MZmine 2.20[66] was used. Acquired mass spectrometry data from the samples was imported in MZmine 2.20 and the total ion current chromatographic data were evaluated. Based on the evaluation, mass detection, chromatogram building, and chromatogram deconvolution was performed in a step-wise manner using the available functionalities in the software. The detected and deconvoluted peaklists containing mass features for each sample were aligned using the RANSAC aligner available in MZmine. The aligned peaklists were exported in.csv format for subsequent chemometric analysis.

Chemometric analysis was performed using MetaboAnalyst 3.0[67]. Before applying chemometrics, the uploaded data were filtered and normalized. Thereafter, differences of ion abundances within the samples were investigated by applying Partial Least Square Discriminate Analysis. To visualize the degree of relatedness among different samples, hierarchical clustering was performed using complete linkage and Euclidean distance. Dendrograms were constructed using the *stats* package in R[68] (version 3.3.3). To generate the distance matrix and dendrogram, the resulting peaklists exported from MZmine were averaged over the four replicates for each sample, giving an average peaklist per sample.

**Nematode reproduction experiment**. Sterilized background soil was prepared as described above. Forty 1 liter pots were filled with 830 g of sterilized background soil and were assigned to one of the eight plant species. After planting of single seedlings per pots, the pots were placed in a randomized block design under the same greenhouse conditions as described above. After 2 weeks of plant growth, a suspension containing ~400 *M. hapla* juveniles was inoculated near the main root of each of the plants. The same watering regime was applied as in the feedback experiment. After 12 weeks, shoots were clipped and dried, and root systems were carefully separated from the soil. All soil from each pot was individually bagged and stored at 4 °C until nematode extraction. Nematodes were subsequently extracted using an Oostenbrink elutriator[69] and concentrated to 10 ml. Subsequently, we extracted nematodes from the roots. For this, roots from all plants were separated in two parts, which both were weighed fresh. One part of the roots then was dried at 70 °C until constant weight, whereas the other half was cut into pieces of 1–2 cm and placed in a mistifier for 4 weeks to extract nematodes from the inside of the roots[69]. Nematode suspensions were harvested from the mistifier after 2 and 4 weeks, combined, and concentrated to 10 ml. Both nematode samples were then counted using an inverse light microscope (×200; Olympus CK40)). Using the total fresh weight and the dry-fresh root weight ratio, total nematode numbers inside the roots were estimated.

**Statistical analyses**. Variations in prokaryotic and eukaryotic communities were explored by running separate Principal component analyses (PCA) in Canoco 5[70], comparing the communities between plant origins and plant species, while including soil replicate as a covariate. We then performed partial redundancy analyses (RDA) to individually test the effect of plant origin and plant species on variation in prokaryotic and eukaryotic communities, while partialling out the variation caused by the different soils. Similar analyses were performed to test plant origin and species effects on variation in the major subgroups of the eukaryotic communities: fungi, protists, and nematodes. We then used vegdist in the R *vegan* package[71] to calculate pairwise community dissimilarities of prokaryotes and all eukaryotes and the eukaryotic subgroups fungi, protists, and nematodes between all eight plant species in each independent soil. Overall, pairwise community dissimilarities were calculated by averaging the pairwise dissimilarities in the five independent soil replicates.

We examined the phylogenetic effects on community composition by testing the correlation between pairwise phylogenetic distances and community dissimilarities using two-tailed Mantel tests in *vegan*, with correlation method pearson and 999 permutations. To determine whether closely related species had more similar root traits than distantly related species, we similarly tested the correlations between pairwise phylogenetic distances and absolute differences in specific root length, average root diameter, and chemical dissimilarity based on the DART analysis. Subsequently, also the correlations between rhizosphere community dissimilarities and trait dissimilarities were tested in a similar way. Based on these correlation tests, we constructed a structural equation model (piecewiseSEM in R) testing the predictive effects of specific root length and root chemistry dissimilarities on community dissimilarities of bacteria, fungi, protists, and nematodes, and the predictive effects of bacterial and fungal community dissimilarities on community dissimilarity of protists and nematodes.

Plant–soil feedback variation among plant species was tested by modeling the biomass response in a general linear model including fixed factors block, soil treatment and plant species, and the plant species*treatment interaction (lm in R). A significant plant species*treatment interaction would indicate that plant species differ in their biomass response to soil conditioning. Overall, feedback differences between native and range-expanding plant species were tested by the specification of a contrast. Significant differences in biomass in the conditioning and control treatments were tested using *lsmeans* (package). Average plant–soil feedback values per plant species were calculated by averaging the feedback value (ln(biomass$_{conditioned}$/biomass$_{control}$)) in each of the five independent soil replicates. To test whether feedback differences were stronger between more distantly related species than between closely related species, we tested the correlation between pairwise phylogenetic distance and pairwise feedback differences using a Mantel test (correlation method Pearson, 999 permutations, two-tailed). Moreover, correlations between plant–soil feedback outcome and the relative abundance of root-feeding nematodes (genera), plant pathogens (genera/families), and AMF were tested to examine whether these groups may have determined the observed plant–soil feedback patterns.

The reproduction of *M. hapla* was modeled using a generalized linear model with a negative binomial distribution[72] that included the fixed factors species and soil replicate (glm.nb in *mass*[73]). Between-species differences were tested using post-hoc Wald's tests with the package *phia*[74]. Finally, we tested the correlation between *Meloidogyne* numbers and plant–soil feedback using a Pearson's correlation test (two-tailed).

**Reporting summary**. Further information on experimental design is available in the Nature Research Reporting Summary linked to this article.

## Data availability
The data that support the findings of this study are available from the corresponding author upon request. Newly sequenced *rbcl* gene information is available at NCBI under the accession numbers MK542498-MK542505. rDNA sequence data of 16S and 18S rRNA gene reads is uploaded to https://www.ebi.ac.uk/ena/data/view/PRJEB29769. Source data for Figs. 2, 3, 4 and 5 are provided with the paper.

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

## Acknowledgements

We thank Carolin Weser, Sneha Gulati, and Mattias de Hollander for technical support, Paul Hoekstra for collection of *G. purpureum* seeds, Madhav Thakur for help with analyses, and Nicky Lustenhouwer, Madhav Thakur, and Koen Verhoeven for commenting on previous versions of the manuscript. This study was supported by the European Research Council (ERC advanced grant ERCAdv 323020 (SPECIALS) to W.H. v.d.P.). This is publication 6680 of the Netherlands Institute of Ecology (NIOO-KNAW).

## Author contributions

R.A.W., S.G., and W.H.v.d.P. designed the study. R.A.W., W.K., and S.G. performed the greenhouse experiments. H.M., P.H., S.G., and R.A.W. performed the molecular lab work. P.G. and P.K. analyzed the metabolomics data. R.A.W. performed the statistical analyses. The manuscript was written by R.A.W., S.G., and W.H.v.d.P., and all authors commented on the manuscript.

## Additional information

**Competing interests:** The authors declare no competing interests.

