## [Peer Review File · Nature Communications]

Reviewers' comments:

Reviewer #1 (Remarks to the Author):

This study investigates plant-soil feedback among four expanding and four native plant species, based on a pot experiment with 80 pots (40 controls and 40 pots to test plant-soil feedback after soil conditioning). It was tested whether phylogenetic distance, root microbiome variation and structural root traits explain plant soil feedback.

So far most plant-soil feedback studies consider soils as a black box. This study moves one step further and identifies potential mechanisms responsible for plant-soil-feedback, which I found interesting and relevant.

While I think this is worthwhile work, I do have several comments.

- 1) This study shows that plant-soil feedback was not explained by overall changes in bacterial, fungal or protist communities. Instead, specific soil functional groups (e.g. nematodes, pathogens and arbuscular mycorrhizal fungi) appeared to be responsible for the observed effects. I would focus more on this and also highlight this observation clearly in the abstract.
- 2) The majority of the results are based on the 40 soil conditioning pots (e.g. 8 plant species x 5 replicates). However, when calculating plant soil feedback (e.g. see figure 4) the average value per plant species is used (resulting in only 1 value per plant species). Would it be possible to provide data for all replicates and perform a nested analysis (e.g. it appears to me the power of the experiment/analysis is not fully explored). This could be done by using the average value of the unconditioned soil and specific values for the conditioned soil. Also, 5 different soils were used (for each replicate a different soil) and simply averaging them appears to be inappropriate (because these are different soils). Obviously, when doing this, the different plant species should then have different symbols (e.g. 40 data points in total).
- 3) In the supplement important additional information is provided. However, important details on the sequencing analysis and data quality are missing. Please provide rarefaction curves for each replicate (number of sequences versus richness) for bacteria, fungi and protists separately (also for the replicates which were removed from analysis). Samples with fewer than 3000 18S rDNA reads were removed (see line 330). Are these processed reads or raw reads? Was this step necessary? How do the results look like if 1000 reads are taken as cut-off value? Where all treatments affected in the same way (how many replicates are present per treatment for sequencing analysis, please show such data in the Supplement)?

other comments:

- 1) Line 123: "whereas differences in bacterial community composition appeared to correspond with variation in root metabolic profiles (Fig. 2)." Note: however, this is not significant and hence it is probably better not to focus on this.
- 2) Would it be possible to show a scatterplot of the correlation between protists and bacteria? This appears to be an interesting observation. Is this usually observed or is this a novel observation?
- 3) Line 329: "Prior to the analyses, we deleted all OTUs present in less than 25% of the samples." Was any of these OTU taxa abundant (e.g. they could be specific to a particular treatment)? Please provide information about this and include a statement like (if true and possible of course): e.g. none of these OTUs were abundant and each contributed less than 0.01% to the total amount of reads.
- 4) Figure 4: the results show that plant soil feedback is negatively correlated with root feeding nematode abundance. This is an appealing result. Which nematode taxa were detected and are

these typically seen as plant parasitic (the focus is now on Meloidogyne)? Is it possible to rank these nematode taxa based on indicator values for their pathogenicity?

5) The results with the pathogens and arbuscular mycorrhizal fungi (Figure 4) are also very interesting (even though not significant). Would it be possible to provide data for all replicates (per plant species) and perform a nested analysis (e.g. it appears to me the power of the experiment/analysis is not fully explored. This might provide further interesting insights, especially because there should be natural variation as each replicate contains a different soil.

6) Fig S4: I counted approximately 25 data points for the graphs. However, the number of pots is 40. Please explain the difference? Does this mean that approximately 15 data points are missing (or have been removed)? This appears to be a lot.

7) Figure 5 shows that there are two plant species with a strong negative plant soil feedback (in relation to nematode abundance). However, in figure 4, three plant species have a strong negative soil feedback in relation to nematode abundance. Are the same plant species negatively affect? Please provide codes for the respective plant species in these figures. What is the reason for the difference?

8) Line 334: "OTUs were manually assigned into functional groups." Would it be possible to show which taxa were assigned to which groups as supplementary information (e.g. perhaps showing the 20 most abundant taxa per group (root feeding nematodes, pathogens and arbuscular mycorrhizal fungi) if many different taxa were detected).

9) Some scientists might argue that correlations based on relative abundance of reads (e.g. figure 4) is not appropriate. I find this result very interesting and relevant.

10) The observed plant-soil feedback values are very low (-1 to -3). This is logical because I assume nutrient availability in the pots without soil conditioning was much higher (because no plants grew in those pots and acquired nutrients, while nutrients were already acquired from the soil in the soils that were conditioned. This should be mentioned in the discussion.

Reviewer #2 (Remarks to the Author):

The authors present an elegant experimental design to tease out the relative roles of species identity, relatedness, and invasiveness in plant-soil feedback (PSF) responses, with an unusually thorough treatment of the influence of these factors and species traits on the "black box" of soil biotic communities. The experimental design and full mechanistic pathway explored are novel and the results likely to advance the field of PSF research by suggesting further focus on particular mechanistic pathways. However, I am not convinced that the analysis fully supports the conclusions made, nor that it currently makes full use of this rich data set. Furthermore, I think that some of the authors' decisions and claims require further justification or explanation.

First, choosing to focus on several species within a single genus is rare when testing the role of phylogenetic relatedness, but the Introduction is currently unclear as to why "deeply conserved traits, which show variation on higher taxonomic levels such as plant families" (Lines 70-71) might be less important to explaining PSF or its role in structuring communities, than are less conserved traits. More clearly setting the stage for the point made in the Discussion in Lines 233-235 would better emphasize the contribution of this paper. On the other hand, if phylogenetic distance across broadly varying community members has been shown to be a useful predictor in other studies, then how is a lack of effect for species that tend not to co-occur helpful for understanding real communities?

Second, could the different components of the experiment be analyzed concurrently to produce an integrated picture of how the predictors both directly and indirectly influence the responses? Some form of structural equation modelling, for example, where Fig. 2 shows the outcome of a path analysis (or mediator analysis, or RLQ analysis to compare more than 2 distance matrices simultaneously). This type of approach could help test the hypothesized mechanistic pathway whereby species/phylogeny/invasiveness influence root traits, which in turn influence soil microbiomes, which in turn influence PSF. As it stands, several of these pairwise links are shown to be significant but if they do not ultimately influence PSF, then are there any expected consequences for plant communities? At the very least, there appear to be many correlation analyses performed (often on the same data, but different subsets) without any correction for the increasing potential of false positive results, which suggests that some of these trends might not be significant.

Third, are results for *Geranium* spp. likely to be broadly applicable to other (diverse, herbaceous) genera? Are *Geranium* spp. more or less likely to be susceptible to root-feeding nematodes, for example? And how common are specialist vs. generalist enemies likely to be among different plant groups, to enable prediction of when phylogenetic relationships are more or less likely to be useful predictors? Finally, might the lack of effect of "invasiveness" in your data be an artefact of the general weedy nature of *Geranium* spp., given that the 4 "native" species are known to naturalize elsewhere in the world (at least 3 of them on multiple continents)?

Fourth, you measured root traits and chemistry in sterile soil. While valuable for examining differences among species, is it likely that these traits would be expressed differently in different types of live soil? For example, some of Jim Bever's work, among others, uses root:shoot ratio as a PSF response, indicating that root biomass, at least, can change with soil biota.

Additional minor comments and questions:

Should the Pearson correlation coefficients be reported as "r"-values rather than R-squared (which I would expect to be the coefficient of determination from a regression model)?

Have you done an analysis for root traits other than biomass affecting root-feeding nematodes to support your statement on Lines 172-173?

Two of the graphs in Fig. 4 appear to be missing their trendlines.

Line 2010: "Janzen-Connell", rather than "Cornell"

It may be helpful to the reader to indicate the directions of correlations (or lack thereof) in descriptions of results. For example, the caption for Fig. S6 indicates "The correlation", but there is no trendline or statistic provided, suggesting this graph shows a lack of correlation.

Reviewers' comments:

Reviewer #1 (Remarks to the Author):

This study investigates plant-soil feedback among four expanding and four native plant species, based on a pot experiment with 80 pots (40 controls and 40 pots to test plant-soil feedback after soil conditioning). It was tested whether phylogenetic distance, root microbiome variation and structural root traits explain plant soil feedback.

So far most plant-soil feedback studies consider soils as a black box. This study moves one step further and identifies potential mechanisms responsible for plant-soil-feedback, which I found interesting and relevant.

Thank you for these nice words and the useful comments on our manuscript. We believe that our study system, with eight closely related species, made it possible to identify these potential mechanisms underlying plant-soil feedback.

While I think this is worthwhile work, I do have several comments.

1) This study shows that plant-soil feedback was not explained by overall changes in bacterial, fungal or protist communities. Instead, specific soil functional groups (e.g. nematodes, pathogens and arbuscular mycorrhizal fungi) appeared to be responsible for the observed effects. I would focus more on this and also highlight this observation clearly in the abstract.

We agree that specific groups, rather than community composition, explain the observed effects. The title of our paper is therefore focussed on the potential importance of nematodes in predicting plant-soil feedbacks: "Root traits and belowground herbivores explain plant-soil feedback variation among congeners". We now added that rhizosphere community dissimilarity did not predict plant-soil feedback differences in the abstract (L36-37). In the discussion, we emphasized this finding more strongly by highlighting the role of nematodes as predictors of plant-soil feedback at the beginning of the second paragraph (L218-220). We therefore moved the sentence on the correlation between bacterial and protist communities to the end of the first paragraph. Also in the last part of the discussion (L269) we now emphasize that root-feeding nematode numbers, and not rhizosphere community dissimilarity best predicted plant-soil feedbacks.

2) The majority of the results are based on the 40 soil conditioning pots (e.g. 8 plant species x 5 replicates). However, when calculating plant soil feedback (e.g. see figure 4) the average value per plant species is used (resulting in only 1 value per plant species). Would it be possible to provide data for all replicates and perform a nested analysis (e.g. it appears to me the power of the experiment/analysis is not fully explored). This could be done by using the average value of the unconditioned soil and specific values for the conditioned soil. Also, 5 different soils were used (for each replicate a different soil) and simply averaging them appears to be inappropriate (because these are different soils). Obviously, when doing this, the different plant species should then have different symbols (e.g. 40 data points in total).

According to the suggestion we now included the same analyses (correlations between plant-soil feedback and functional groups of biota in the rhizosphere) using all individual replicates, colour-coded by species. In this analysis, we calculated individual plant-soil feedbacks using biomass in the conditioned soil and biomass in control soils of the same origin, so that plant-soil feedback measures

are always based on a plant growing in the same (conditioned or unconditioned) replicate soil. Calculating plant-soil feedbacks using the average of plant responses in the control soils would have strongly decreased the independence of the replicate measures, as well as artificially reduced the variation(1).

We included this new analysis as Supplementary figure 9. This new analysis shows the same pattern for root-feeding nematodes and a similar outcome of the correlation test ($p < 0.05$). However, as expected, there is a lot of scatter around this trend, which is explained by the use of 5 fully independent soil replicates. We cannot perform a nested analysis, because we did not replicate plant species on the individual soils. The main purpose of our experiment was to examine generalizable plant-soil feedback differences between 8 Geranium species, while differences between soils were not the target of our study. Therefore we did not replicate different soils and used soils from different sampling locations as replicates. Consequently these soils differ in their composition of soil organisms (including nematodes), creating considerable variation. We prefer to keep the analysis based on average values as the main figure, because this more clearly shows the overall differences between plant species.

3) In the supplement important additional information is provided. However, important details on the sequencing analysis and data quality are missing. Please provide rarefaction curves for each replicate (number of sequences versus richness) for bacteria, fungi and protists separately (also for the replicates which were removed from analysis). Samples with fewer than 3000 18S rDNA reads were removed (see line 330). Are these processed reads or raw reads? Was this step necessary? How do the results look like if 1000 reads are taken as cut-off value? Where all treatments affected in the same way (how many replicates are present per treatment for sequencing analysis, please show such data in the Supplement)?

As suggested, we added additional information to the supplementary files. We calculated rarefaction curves for each sample bacteria, protists, fungi and nematodes separately (Supplementary figure 3).

We decided to apply a cutoff of 3000 18S rRNA gene reads as our aim was to obtain diversity information on protists, fungi and nematodes. As nematode reads represented on average 3.7% of all 18S rRNA gene reads we aimed for more than 100 nematode reads in all samples in accordance to the general rule for morphological investigations to identify a minimum of 100-150 specimen, as that enables to obtain reliable nematode community information(2).

Even with a cutoff of 3000 (processed) reads we only had to remove 2 samples from the 18S dataset, whereas we had to remove 1 sample if we had used a cutoff of 1000. We added an overview of the remaining samples (Supplementary table 5), showing that a maximum of one replicate had to be removed from individual treatments (plant species).

other comments:

1) Line 123: “whereas differences in bacterial community composition appeared to correspond with variation in root metabolic profiles (Fig. 2).” Note: however, this is not significant and hence it is probably better not to focus on this.

We agree this should not be the focus, however, as the correlation coefficient was rather substantial ($r = 0.52$), we think that the trend may need to be mentioned in the text. In order to acknowledge that it is a trend only, we now replaced 'appeared to' with 'tended' (L131).

As proposed by reviewer 2, we now included a structural equation model, testing predictive effects between root traits and rhizosphere community groups (Supplementary figure 7). This model confirms the predictive effect of root chemistry on bacterial community composition, and hence we address this link in the discussion (L213-214).

2) Would it be possible to show a scatterplot of the correlation between protists and bacteria? This appears to be an interesting observation. Is this usually observed or is this a novel observation?

We now added scatterplots showing the relationships (or the lack thereof) between the community dissimilarities of the different organismal groups (protists, nematodes, bacteria, fungi) as Supplementary Figure 6.

Due to the known direct predator-prey relationships between protists and bacteria(3, 4) it is likely that communities of protists and bacteria are affecting each other. Protists may feed on their bacterial prey in a species-specific way(5, 6). Yet, it is striking that the correlation between entire protist and bacterial community dissimilarities is significant, suggesting that specialization in these predator-prey relationships may be a driver of protist and bacterial community structure. This correlation was also confirmed by the structural equation model (Supplementary Figure 7). We now emphasized this finding more strongly in the manuscript (L214-217).

3) Line 329: "Prior to the analyses, we deleted all OTUs present in less than 25% of the samples." Was any of these OTU taxa abundant (e.g. they could be specific to a particular treatment)? Please provide information about this and include a statement like (if true and possible of course): e.g. none of these OTU's were abundant and each contributed less than 0.01???% to the total amount of reads.

We took this highly conservative approach of reducing taxa in order to decrease sampling biases induced by including rare taxa. Some of the OTU's that were removed had more than 1000 reads throughout the database, but these OTU's were not specific to a certain treatment. Most of these OTUs represented larger animal taxa (e.g. *Drosophila* and *Enchytraeus*), which were abundant in only a few samples.

We included the following sentence to better describe our approach to focus on representative OTUs: L353-355 "While a small number of these removed OTUs had high abundance in some samples, none of them were consistently abundant in any of the treatments."

4) Figure 4: the results show that plant soil feedback is negatively correlated with root feeding nematode abundance. This is an appealing result. Which nematode taxa were detected and are these typically seen as plant parasitic (the focus is now on Meloidogyne)? Is it possible to rank these nematode taxa based on indicator values for their pathogenicity?

It is not possible to assign pathogenicity levels to individual nematode taxa as their individual effects on the tested plant species are not known, but we now included the detected root-feeding

nematode taxa in a table where we show the functional group assignment (Supplementary Table 6). *Meloidogyne* dominated root-feeding nematode communities (representing on average 68% of all root-feeding nematodes L192), and therefore we focussed on this genus. Overall, *Meloidogyne* species are considered as most important nematode pests in agriculture(7), which we emphasized in the discussion (220-221).

5) The results with the pathogens and arbuscular mycorrhizal fungi (Figure 4) are also very interesting (even though not significant). Would it be possible to provide data for all replicates (per plant species) and perform a nested analysis (e.g. it appears to me the power of the experiment/analysis is not fully explored. This might provide further interesting insights, especially because there should be natural variation as each replicate contains a different soil.

We included the correlations between plant-soil feedback and pathogens, as well as arbuscular mycorrhizal fungi using all individual replicates (Supplementary Figure. 9). Also in those analyses the correlations are not statistically significant. As explained in response to your second major comment, we cannot perform a nested analysis.

6) Fig S4: I counted approximately 25 data points for the graphs. However, the number of pots is 40. Please explain the difference? Does this mean that approximately 15 data points are missing (or have been removed)? This appears to be a lot.

The panels in Figure S4 do not show data points on the pot level, but show 28 independent pairwise averaged root trait differences correlated with pairwise phylogenetic distance. All of these data points are present, although some dots overlap.

7) Figure 5 shows that there are two plant species with a strong negative plant soil feedback (in relation to nematode abundance). However, in figure 4, three plant species have a strong negative soil feedback in relation to nematode abundance. Are the same plant species negatively affect? Please provide codes for the respective plant species in these figures. What is the reason for the difference?

We now colour-coded both figures (4 & 5) to enable comparison. Reproduction of *Meloidogyne* was low on *G. lucidum* whereas this species accumulated relatively high numbers of root-feeding nematodes in the conditioning experiment, explaining why there are two plant species with high nematode numbers turning up in this analysis.

While carefully checking the graphs, we noticed that nematode reproduction data of one pair of species were switched, which we now corrected in the updated graph (Fig. 5). This switch did not change the statistical outcome of correlation test.

8) Line 334: "OTUs were manually assigned into functional groups." Would it be possible to show which taxa were assigned to which groups as supplementary information (e.g. perhaps showing the 20 most abundant taxa per group (root feeding nematodes, pathogens and arbuscular mycorrhizal fungi) if many different taxa were detected).

We now added a table to the supplementary information (Supplementary Table 6) showing which taxa were assigned to the following functional groups: root-feeding nematodes, plant pathogens and arbuscular mycorrhizal fungi.

9) Some scientists might argue that correlations based on relative abundance of reads (e.g. figure 4) is not appropriate. I find this result very interesting and relevant.

We agree and are aware of this as sometimes relative abundances and absolute abundances differ in their effects(8). However, our results are based on both relative (based on sequencing results) and absolute abundances based on the nematode reproduction experiment. This strengthens the observed correlation patterns between root-feeding nematode abundance and plant-soil feedback strength.

10) The observed plant-soil feedback values are very low (-1 to -3). This is logical because I assume nutrient availability in the pots without soil conditioning was much higher (because no plants grew in those pots and acquired nutrients, while nutrients were already acquired from the soil in the soils that were conditioned. This should be mentioned in the discussion.

We agree that nutrient availability may –at least in part - explain the negative feedbacks in our experiment. However, we did supply Hoagland nutrient solution in the feedback phase to all mesocosms, so that there should not have been limitation of macro-nutrients (L321-322). Moreover, plant biomass in the conditioning phase did not correlate with the strength of the negative plant-soil feedbacks (see Figure below). Nevertheless, some nutrients may still have been limited in the feedback phase, and we added to the manuscript (L230-233): “Apart from the biotic actors, the observed negative plant-soil feedbacks may to some degree have been caused by differences in nutrient availability between the control and conditioned pots. While we added Hoagland solution to compensate for the lack of macro-nutrients, it cannot be excluded that plants in the feedback phase still suffered from shortage of certain micronutrients.”

Reviewer #2 (Remarks to the Author):

The authors present an elegant experimental design to tease out the relative roles of species identity, relatedness, and invasiveness in plant-soil feedback (PSF) responses, with an unusually thorough treatment of the influence of these factors and species traits on the "black box" of soil biotic communities. The experimental design and full mechanistic pathway explored are novel and the results likely to advance the field of PSF research by suggesting further focus on particular

mechanistic pathways. However, I am not convinced that the analysis fully supports the conclusions made, nor that it currently makes full use of this rich data set. Furthermore, I think that some of the authors' decisions and claims require further justification or explanation.

Thank your positive remarks and the helpful comments.

First, choosing to focus on several species within a single genus is rare when testing the role of phylogenetic relatedness, but the Introduction is currently unclear as to why "deeply conserved traits, which show variation on higher taxonomic levels such as plant families" (Lines 70-71) might be less important to explaining PSF or its role in structuring communities, than are less conserved traits. More clearly setting the stage for the point made in the Discussion in Lines 233-235 would better emphasize the contribution of this paper. On the other hand, if phylogenetic distance across broadly varying community members has been shown to be a useful predictor in other studies, then how is a lack of effect for species that tend not to co-occur helpful for understanding real communities?

While studies including broadly varying community members examine whether there is an overall phylogenetic pattern in plant-soil feedback variation among the community members, they do not address the question whether the most closely related congeners (e.g. sister taxa) have the most comparable plant-soil feedbacks. We think this a relevant question, as a confirmation of such a pattern would increase the predictability of plant-soil feedbacks outcomes of individual species. We clarified this, by adding the following sentence (L71-73): "Therefore, such studies may not be suitable to test whether phylogenetic distances can be used to understand variation in plant-soil feedbacks among closely related congeneric species."

With our study, we do not aim to make statements on the level of communities, but instead want to explain how closely related species differ in plant-soil feedback patterns, and which mechanisms underlie this variation. We agree that phylogenetic distances also help to explain such variation, but not at small phylogenetic scales. We therefore added in the discussion (L252-255): "Thus, to fully understand plant-soil feedback variation among more closely and more distantly related plant species, a combination of phylogenetically conserved traits and un-conserved traits is needed."

Second, could the different components of the experiment be analyzed concurrently to produce an integrated picture of how the predictors both directly and indirectly influence the responses? Some form of structural equation modelling, for example, where Fig. 2 shows the outcome of a path analysis (or mediator analysis, or RLQ analysis to compare more than 2 distance matrices simultaneously). This type of approach could help test the hypothesized mechanistic pathway whereby species/phylogeny/invasiveness influence root traits, which in turn influence soil microbiomes, which in turn influence PSF.

We followed the suggestion and applied structural equation modelling to explore how root trait variation is affecting community dissimilarity of bacteria and fungi, and their consumers – protists and nematodes, thereby expanding the (correlation) analyses underlying Figure 2. The results of this SEM analysis are shown in Supplementary Figure 7, and described in the results (L139-144) and methods sections (L437-440). Furthermore, we highlighted the significant links, obtained from the structural equation model, in Figure 2.

We did not include plant-soil feedback in these analyses, as our earlier analyses showed that rhizosphere community dissimilarity does not explain plant-soil feedback variation (Supplementary Figure 8).

As it stands, several of these pairwise links are shown to be significant but if they do not ultimately influence PSF, then are there any expected consequences for plant communities?

One of the main goals of our study was to increase the mechanistic understanding on the determinants of root microbiomes, including nematodes. We show that microbiome variation indeed does not necessarily affect plant-soil feedback, as only some components of the microbiome (e.g. root-feeding nematodes) may be determining plant-soil feedback effects on conspecifics. Our study was not set-up to answer questions on the level of plant communities, but focusses on the performance of individual plant species in response to their own soil community. As such we cannot address the question on consequences for plant communities.

At the very least, there appear to be many correlation analyses performed (often on the same data, but different subsets) without any correction for the increasing potential of false positive results, which suggests that some of these trends might not be significant.

We agree that carrying out many correlation analyses is prone to false positive results, and indeed some of the (weakly) significant correlations may not be indicative of an ecological mechanism. Nevertheless, we used correlation tests to explore possible links in our data. We now added the structural equation model to further test the observed correlations, and this confirms the relations that we focussed on most strongly: 1) root chemistry – bacteria, 2) specific root length – fungi, and 3) bacteria – protists. Furthermore, we set up the nematode reproduction experiment to experimentally verify whether the correlation between plant-soil feedback and root-feeding nematode numbers also holds in terms for absolute abundances of root-feeding nematodes.

Third, are results for *Geranium* spp. likely to be broadly applicable to other (diverse, herbaceous) genera? Are *Geranium* spp. more or less likely to be susceptible to root-feeding nematodes, for example? And how common are specialist vs. generalist enemies likely to be among different plant groups, to enable prediction of when phylogenetic relationships are more or less likely to be useful predictors? Finally, might the lack of effect of "invasiveness" in your data be an artefact of the general weedy nature of *Geranium* spp., given that the 4 "native" species are known to naturalize elsewhere in the world (at least 3 of them on multiple continents)?

We think that our results are likely to be broadly applicable to other groups of herbaceous plant species. One of our earlier studies showed that *Geranium* species are not an outlier in accumulation of root-feeding nematodes(9).

Whether the ratios of specialists and generalists differ between different groups of plant species is a very interesting point. However, we think this is impossible to target because of a lack of knowledge on which belowground natural enemies are true specialists. In the case of nematodes, there are a few examples of nematode species that have a phylogenetically limited host range, for example the potato cyst nematode *Globodera rostochiensis*(10).

We don't think the lack of effect of plant origin is an artefact of the weedy nature of *Geranium* species. Also (and perhaps especially) 'weedy' species have been shown to experience enemy release(11), so in their new range non-native weeds may have a benefit over native weeds. Yet, in our study this was clearly not the case.

Fourth, you measured root traits and chemistry in sterile soil. While valuable for examining differences among species, is it likely that these traits would be expressed differently in different

types of live soil? For example, some of Jim Bever's work, among others, uses root:shoot ratio as a PSF response, indicating that root biomass, at least, can change with soil biota.

We agree that plant traits may differ between live soil and sterilized soil. However, we want to emphasize that the conditioning phase soils consisted out of 90% sterilized soil, inoculated with 10% live soil. Plants were therefore initially growing in soil with reduced abundance of (plant-pathogenic) organisms. During plant growth, microbes will have enumerated on the developing plant roots. Then, in the feedback phase, the growing plants will be maximally exposed to a more fully developed microbiome. We therefore believe that in the conditioning phase the traits had a stronger effect on the soil community than vice versa.

Additional minor comments and questions:

Should the Pearson correlation coefficients be reported as "r"-values rather than R-squared (which I would expect to be the coefficient of determination from a regression model)?

We agree this is more informative, and changed this throughout the manuscript and figures.

Have you done an analysis for root traits other than biomass affecting root-feeding nematodes to support your statement on Lines 172-173?

We tested the correlation between plant biomass and root-feeding nematodes to exclude the possibility that nematode numbers would be simply highest on the plants with the biggest roots. The absence of such a correlation shows that nematode numbers are predicted by other traits than just pure root (= resource) quantity. We checked whether other structural root traits (average root diameter, specific root length, root/shoot ratio) were related to nematode numbers, but this was not the case. Earlier work showed that species-specific root chemistry can be an important determinant for nematode performance(9), but because we performed untargeted root chemical profiling, it is impossible to reliably link chemistry to root-feeding nematode performance in the present study.

Two of the graphs in Fig. 4 appear to be missing their trendlines.

We initially did not add trend lines because of the non-significant correlation test, but now added them.

Line 2010: "Janzen-Connell", rather than "Cornell"

We changed this typo in the manuscript.

It may be helpful to the reader to indicate the directions of correlations (or lack thereof) in descriptions of results. For example, the caption for Fig. S6 indicates "The correlation", but there is no trendline or statistic provided, suggesting this graph shows a lack of correlation.

We now added trendlines and statistics to all scatterplots in the manuscript and supplements and updated the captions.

References used

1. Brinkman EP, Van der Putten WH, Bakker EJ, & Verhoeven KJ (2010) Plant–soil feedback: experimental approaches, statistical analyses and ecological interpretations. *J. Ecol.* 98(5):1063-1073.
2. Wiesel L, Daniell TJ, King D, & Neilson R (2015) Determination of the optimal soil sample size to accurately characterise nematode communities in soil. *Soil Biol. Biochem.* 80(0):89-91.
3. Geisen S, *et al.* (2018) Soil protists: a fertile frontier in soil biology research. *FEMS Microbiology Reviews* 42(3):293-323.
4. Saleem M, Fetzer I, Dormann CF, Harms H, & Chatzinotas A (2012) Predator richness increases the effect of prey diversity on prey yield. *Nature Communications* 3:1305.
5. Rønn R, McCaig AE, Griffiths BS, & Prosser JI (2002) Impact of Protozoan Grazing on Bacterial Community Structure in Soil Microcosms. *Applied and Environmental Microbiology* 68(12):6094-6105.
6. Schulz-Bohm K, *et al.* (2016) The prey's scent – Volatile organic compound mediated interactions between soil bacteria and their protist predators. *The ISME Journal* 11:817.
7. Jones JT, *et al.* (2013) Top 10 plant-parasitic nematodes in molecular plant pathology. *Molecular Plant Pathology* 14(9):946-961.
8. Geisen S, *et al.* (2018) Integrating quantitative morphological and qualitative molecular methods to analyse soil nematode community responses to plant range expansion. *Methods in Ecology and Evolution* 9(6):1366-1378.
9. Wilschut RA, Silva JCP, Garbeva P, & van der Putten WH (2017) Belowground Plant–Herbivore Interactions Vary among Climate-Driven Range-Expanding Plant Species with Different Degrees of Novel Chemistry. *Frontiers in Plant Science* 8(1861).
10. Kort J, Ross H, Rumpfenhorst H, & Stone A (1977) An international scheme for identifying and classifying pathotypes of potato cyst-nematodes *Globodera rostochiensis* and *G. pallida*. *Nematologica* 23(3):333-339.
11. Blumenthal D, Mitchell CE, Pyšek P, & Jarošík V (2009) Synergy between pathogen release and resource availability in plant invasion. *Proceedings of the National Academy of Sciences* 106(19):7899-7904.

Reviewers' comments:

Reviewer #1 (Remarks to the Author):

The authors addressed my comments and added additional information in the Supplement.

Supplementary figure 9: This is an extended version of figure 4. It shows now that even within plant species (among replicates) there is high variation in plant soil feedback (e.g. it indicates that plant soil feedback depends on the interaction between plant species and soil type). However, with the current experimental design it is not possible to test whether such an interaction term exists and the figure shows that a lot of variation is unexplained.

The authors explanation of averaging per plant species makes sense based on the research question (e.g testing for plant-soil feedback differences among plant species). However, this explanation is rather plant focused and the presented results indicate that it is questionable whether averaging is correct (e.g. for some plant species, replicates with low and high nematode abundance and weak and strong plant soil feedback are added together – not necessarily (as far as I can see) with a clear causal relationship between nematode abundance and plant soil feedback). Moreover, the calculation of plant soil feedback based on only two pots – the conditioned and unconditioned soil – is highly sensitive to small (random) changes/deviations in one of the pots and this may cause a lot of variation in plant-soil feedback. Is there perhaps a different way to show a relationship between nematode abundance and plant-soil feedback or plant growth (e.g. only focusing only on plant biomass in conditioned soils)? Such data are necessary to substantiate the conclusion (and title) that belowground herbivores explain plant soil feedback. In view of these results the conclusions need to be further softened and in the discussion it is important to mention that further research (with more soils + replicated soils) is necessary to substantiate and generalize these findings.

The sequencing depth is okay (though not perfect, especially not for the nematodes). It would be good to add a qualifier in the text (or in the supplement) stating that enhanced sequencing depth and use of further primers targeting different groups of soil biota may further contribute to the explanatory power of the data-set.

Supplementary figure 8: please check the P-levels.

Reviewer #2 (Remarks to the Author):

As I stated in my previous review, this study provides an important contribution to the plant-soil feedbacks literature both in its thorough exploration of effects of different components of the soil microbiome and in testing the effects of multiple plant attributes (species relatedness and invasiveness) on PSF responses. The authors have addressed my comment about incorporating an additional analysis (SEM), thus strengthening their conclusions.

However, I do think that two of my previous comments about the focal genus selected for this study could be addressed more fully in the paper, especially given the authors' points in their Response to Reviews and the potential for these additions to make their arguments in the Discussion stronger.

First, the authors point out in their Response that there is a lack of knowledge on true specialist belowground enemies, with few examples of nematodes with a phylogenetically-limited host range. This statement includes two points that are likely relevant to the Discussion (Lines 222-225): 1) if root-feeding nematodes are important predictors of PSF and are rarely specialists, and specialists are required for phylogenetic signal of PSF, then relatedness is unlikely to be predictor of PSF (not

just in this case with *Meloidogyne* as the main player in the soil microbiome); and 2) if belowground specialists are necessary for relatedness to predict PSF, but little is known about phylogenetic range of hosts, then understanding more about nematode host range would help us better understand the potential for PSF to change with plant relatedness.

Second, this study finds no effect of plant origin on PSF, which suggests a lack of enemy release for range-expanding species. Given that the "native" species in this experiment are known to have expanded their ranges broadly elsewhere, there may be evidence in the literature of them experiencing enemy release -- but if they are not demonstrated to do so, perhaps that could help explain the lack of origin effect found here. Also, work by Agrawal et al 2005 (<https://doi.org/10.1890/05-0219>) suggests that enemy release is inconsistent from different guilds of enemies -- perhaps the range-expanding species were unable to escape from root-feeding nematodes specifically?

Reviewer #1 (Remarks to the Author):

The authors addressed my comments and added additional information in the Supplement.

Supplementary figure 9: This is an extended version of figure 4. It shows now that even within plant species (among replicates) there is high variation in plant soil feedback (e.g. it indicates that plant soil feedback depends on the interaction between plant species and soil type). However, with the current experimental design it is not possible to test whether such an interaction term exists and the figure shows that a lot of variation is unexplained.

The authors explanation of averaging per plant species makes sense based on the research question (e.g testing for plant-soil feedback differences among plant species). However, this explanation is rather plant focused and the presented results indicate that it is questionable whether averaging is correct (e.g. for some plant species, replicates with low and high nematode abundance and weak and strong plant soil feedback are added together – not necessarily (as far as I can see) with a clear causal relationship between nematode abundance and plant soil feedback). Moreover, the calculation of plant soil feedback based on only two pots – the conditioned and unconditioned soil – is highly sensitive to small (random) changes/deviations in one of the pots and this may cause a lot of variation in plant-soil feedback. Is there perhaps a different way to show a relationship between nematode abundance and plant-soil feedback or plant growth (e.g. only focusing only on plant biomass in conditioned soils)? Such data are necessary to substantiate the conclusion (and title) that belowground herbivores explain plant soil feedback. In view of these results the conclusions need to be further softened and in the discussion it is important to mention that further research (with more soils + replicated soils) is necessary to substantiate and generalize these findings

Answer: We agree that our data cannot be used to test the interaction between plant species and soil type. For that we should have replicated each plant species on each individual soil. Indeed, the replicate feedback values are based on outcomes in two pots: the control pot and the pot with conditioned soil. However, these pots were coupled right from the start of the experiment, so that the variation among replicates represents the natural variation in the field. Correlating nematode numbers with biomass in the conditioned soil is possible, however, as all plant species differ strongly in biomass production, this will drive and determine our correlations. The best way to link nematode numbers with biomass is using proportionalized values for all plant species and the best way to do that is by calculating plant-soil feedback.

We nevertheless agree with the reviewer that the title may be weakened in order to better reflect the nature of our analysis. That is why we revised the title into: Root traits and belowground herbivores relate to plant-soil feedback variation among congeners.

In line with that, we aligned our conclusions in the discussion by changing "can affect" to "likely affect" and added the following sentence saying: "Further research might consider interactions between plant species and soil type, increased numbers of replicates to account for the variation, and inoculation trials in order to experimentally verify the role of plant-feeding nematodes in plant growth reduction."

The sequencing depth is okay (though not perfect, especially not for the nematodes). It would be good to add a qualifier in the text (or in the supplement) stating that enhanced sequencing depth and use of further primers targeting different groups of soil biota may further contribute to the explanatory power of the data-set.

Answer: We agree that the sequencing depth in comparison to studies focusing on bacteria and fungi is comparably low. Yet, considering that nematode communities are usually qualitatively analysed by morphological identification of 100 to 150 species, we believe that the sequencing depth we obtained is sufficiently high to support our ecological interpretations, especially considering that we focused our analyses on functional group level. To acknowledge the potential for higher taxonomic resolution analyses using higher sequencing depth we added the following sentence to the Discussion: Furthermore, deeper sequencing, the use of nematode-specific primers, or longer-read sequencing in future studies should be considered to increase taxon sampling and the taxonomic resolution focusing on root-feeding nematodes as well as other soil-borne pathogens in order to potentially identify taxa that are specifically hosted by the different plant species.

Supplementary figure 8: please check the P-levels.

Answer: We checked this figure and now inserted the p-values from Mantel tests, instead of Pearson correlation tests.

Reviewer #2 (Remarks to the Author):

As I stated in my previous review, this study provides an important contribution to the plant-soil feedbacks literature both in its thorough exploration of effects of different components of the soil microbiome and in testing the effects of multiple plant attributes (species relatedness and invasiveness) on PSF responses. The authors have addressed my comment about incorporating an additional analysis (SEM), thus strengthening their conclusions.

However, I do think that two of my previous comments about the focal genus selected for this study could be addressed more fully in the paper, especially given the authors' points in their Response to Reviews and the potential for these additions to make their arguments in the Discussion stronger.

First, the authors point out in their Response that there is a lack of knowledge on true specialist belowground enemies, with few examples of nematodes with a phylogenetically-limited host range. This statement includes two points that are likely relevant to the Discussion (Lines 222-225): 1) if

root-feeding nematodes are important predictors of PSF and are rarely specialists, and specialists are required for phylogenetic signal of PSF, then relatedness is unlikely to be predictor of PSF (not just in this case with *Meloidogyne* as the main player in the soil microbiome); and 2) if belowground specialists are necessary for relatedness to predict PSF, but little is known about phylogenetic range of hosts, then understanding more about nematode host range would help us better understand the potential for PSF to change with plant relatedness.

Answer: The reviewer raises several interesting aspects. There are also several other possibilities and although we can agree with them, we can also add some more, for example that the various plant species differ in their sensitivity to the nematodes, or that there can be plant species-specific nematode strains (= pathotypes). However, with the current data, any further conclusions would be speculative.

Second, this study finds no effect of plant origin on PSF, which suggests a lack of enemy release for range-expanding species. Given that the "native" species in this experiment are known to have expanded their ranges broadly elsewhere, there may be evidence in the literature of them experiencing enemy release -- but if they are not demonstrated to do so, perhaps that could help explain the lack of origin effect found here. Also, work by Agrawal et al 2005 (<https://doi.org/10.1890/05-0219>) suggests that enemy release is inconsistent from different guilds of enemies -- perhaps the range-expanding species were unable to escape from root-feeding nematodes specifically?

Answer: Most of the plants examined in this study were not yet included in any ecological studies, particularly those investigating enemy release mechanisms, with the exception of the "native" *G. molle* and the range expander *G. pyrenaicum* that we studied before ((Wilschut et al. 2016; Wilschut et al. 2017; Koorem et al. 2018; Wilschut et al. 2018)). Indeed, as suggested, none of these studies showed any clear evidence that enemy release differs between native and range expanding *Geranium* species.

We agree that specific organisms that were not studied could negatively influence plant performance in the new range. However, as we screened a wide variety of bacterial, fungal, protist and nematode taxa, we think that enemy release does not appear to happen among the range expanding *Geranium* plants studied here. One possible reason, which is in fact also suggested by the reviewer, is that the presence of closely related plant species enables enemies to be present in the expanded range that are capable of controlling the range expanding plant species similar to the native *Geranium* species. We now added the following sentence to the conclusions: Enemy-release is likely not underlying the success of *Geranium* species to expand their range as the performance of native and range-expanding plant species was similar.

References

- Koorem, K., Kostenko, O., Snoek, L.B., Weser, C., Ramirez, K.S., Wilschut, R.A. & van der Putten, W.H. (2018) Relatedness with plant species in native community influences ecological consequences of range expansions. *Oikos*, **127**, 981-990.
- Wilschut, R., Geisen, S., Ten Hooven, F. & van der Putten, W. (2016) Interspecific differences in nematode control between range-expanding plant species and their congeneric natives. *Soil Biology and Biochemistry*, **100**, 233-241.
- Wilschut, R.A., Kostenko, O., Koorem, K. & van der Putten, W.H. (2018) Nematode community responses to range-expanding and native plant communities in original and new range soils. *Ecology and Evolution*, **8**, 10288-10297.
- Wilschut, R.A., Silva, J.C.P., Garbeva, P. & van der Putten, W.H. (2017) Belowground Plant–Herbivore Interactions Vary among Climate-Driven Range-Expanding Plant Species with Different Degrees of Novel Chemistry. *Frontiers in Plant Science*, **8**.

REVIEWERS' COMMENTS:

Reviewer #1 (Remarks to the Author):

The authors addressed my comments. I like the approach and concept of the paper. In this experiment plant-soil feedback was assessed for 8 different plant species. In order to measure plant-soil feedback, the 8 plant species were planted in 5 different soils. These 5 different soils acted as individual replicates to measure plant-soil feedback. However, these 5 soils itself were not replicated. The observed large variation in plant soil feedback within treatments (see Figure S9, especially for *G.dissectum* and *G. purpureum*) could, thus, be due to enhanced sensitivity for nematodes (as discussed) or caused by experimental issues and natural variation among replicates of a treatment. It is not possible to completely solve this with the current data-set and experimental design (except repeating the experiment and including additional replicates for the 5 different soils used). Thus, although I like the approach, I am not fully convinced. The evidence presented in figure 5 (e.g. enhanced abundance of nematodes on *G.dissectum* and *G. purpureum*) provides additional support for the findings and conclusions of the paper and tip the balance to a favourable evaluation.

Reviewer #2 (Remarks to the Author):

The authors have addressed reviewer comments and their conclusions and interpretations appropriately reflect the data.

Reviewer #1 (Remarks to the Author):

The authors addressed my comments. I like the approach and concept of the paper. In this experiment plant-soil feedback was assessed for 8 different plant species. In order to measure plant-soil feedback, the 8 plant species were planted in 5 different soils. These 5 different soils acted as individual replicates to measure plant-soil feedback. However, these 5 soils itself were not replicated. The observed large variation in plant soil feedback within treatments (see Figure S9, especially for *G. dissectum* and *G. purpureum*) could, thus, be due to enhanced sensitivity for nematodes (as discussed) or caused by experimental issues and natural variation among replicates of a treatment. It is not possible to completely solve this with the current data-set and experimental design (except repeating the experiment and including additional replicates for the 5 different soils used). Thus, although I like the approach, I am not fully convinced. The evidence presented in figure 5 (e.g. enhanced abundance of nematodes on *G. dissectum* and *G. purpureum* provides additional support for the findings and conclusions of the paper and tip the balance to a favourable evaluation.

Answer: we thank the reviewer for the careful and positive evaluation of our work.

Reviewer #2 (Remarks to the Author):

The authors have addressed reviewer comments and their conclusions and interpretations appropriately reflect the data.

Answer: we also thank reviewer 2 for the careful and positive evaluation of our work.